# Cooperation with Persons with Intellectual Disabilities: Reflections of Co-Researchers Associated with Conducting Inclusive Research

**Katarzyna Ćwirynkało** [1] , **Monika Parchomiuk** [2,*] **and Agnieszka Wołowicz** [3]

[1]   Institute of Pedagogy, Faculty of Social Sciences, University of Warmia and Mazury, 10-710 Olsztyn, Poland; k.cwirynkalo@uwm.edu.pl

[2]   Institute of Pedagogy, Faculty of Pedagogy and Psychology, Maria Curie-Sklodowska University, 20-031 Lublin, Poland

[3]   Faculty of Education, University of Warsaw, 00-927 Warsaw, Poland; a.wolowicz@uw.edu.pl

*   Correspondence: monika.parchomiuk@mail.umcs.pl

**Abstract:** Traditionally, persons with intellectual disabilities in Poland have been researched on and treated as not competent enough to research with. Despite its scientific merit and practical usefulness, inclusive research involving persons with intellectual disabilities as co-researchers has only recently become a subject of interest in this country. The added value of inclusive projects should be analysed from two standpoints, i.e., those of co-researchers with disabilities and co-researchers without disabilities. In this article, we discuss the latter perspective, focusing on Polish researchers without disabilities who have experience in conducting inclusive research with persons with intellectual disabilities. The key aspect of the analyses was to highlight the potential of persons with intellectual disabilities as co-researchers. As a result, we have determined several important aspects of inclusive research in the relational perspective involving co-researchers with intellectual disabilities and co-researchers without intellectual disabilities and the community perspective. The analyses identified four superordinate themes: building relationships in the research team; opportunities and constraints associated with the implementation of inclusive projects; institutional barriers; and the importance of the role of a co-researcher without intellectual disabilities. Implications for further research and practice are discussed.

**Keywords:** inclusive research; intellectual disability; co-researchers; empowerment

## 1. Introduction

### 1.1. Evolution of Inclusive Research

Scientific research should allow for the diversity of a society with respect to its social and cultural characteristics. This means taking into account the perspectives of groups that qualify as diverse not only in terms of age, gender, colour, ethnicity, or cultural and social backgrounds but also in view of different abilities. Intellectual disability is an important but often ignored facet of diversity (Hart et al. 2020). Traditionally, persons with intellectual disabilities have been the subject of research from the perspective of parents, social and support workers, teachers, psychologists, and other professionals (Adderley et al. 2015). Recent years have seen a shift in this approach, reflecting a substantial interest in inclusive research involving persons with intellectual disabilities (Bigby et al. 2014).

The change in the research perspective towards recognising the importance of self-report exploration, which, to some extent, can be associated with the tendencies of emancipation, self-determination, and normalisation, is not free from many methodological and ethical challenges. Even though it is declaratively recognised in scholarly discourse that persons with intellectual disabilities are full-fledged members of society, they are still treated in practice as passive recipients of knowledge, care, and power imposed by

people without disabilities. Adopting the perspective of a human rights-based model of disability means not only acknowledging the agential potential of persons with intellectual disabilities and their expertise in the field of their everyday experience but also embracing the ethical imperative to conduct research in a manner which situates the reflections and experiences of that group in the centre, as opposed to privileging the knowledge of the researcher.

### 1.2. Benefits and Challenges of Inclusive Research

Inclusive research may be an answer to the social exclusion experienced by such individuals and the lack of their full participation in different spheres of life (Kramer et al. 2011). It has been noted that this type of research enables persons with intellectual disabilities to be empowered, to step out of the role of research subjects, and to assume the valued social roles of leaders and research experts (Walmsley 2001; Lloyd et al. 2019; Aubrecht et al. 2021). By being both research participants and co-researchers, persons with intellectual disabilities feel valued and included, experience equality, and improve their self-esteem. They also have the opportunity to take on new challenges, learn new things, and demonstrate their skills (McDonald et al. 2016). This changes the nature of the relationship between researchers without disabilities and persons with intellectual disabilities. They become more democratic, relying on respect and partnership (Lloyd et al. 2019).

There is abundant literature demonstrating the added value of inclusive research projects (Björnsdóttir and Svensdóttir 2008; Bigby et al. 2014; Armstrong et al. 2019). Inclusive research involving persons with intellectual disabilities not only offers better insights into the experiences and opinions of those with intellectual disabilities but also fosters a positive image of persons with intellectual disabilities (McDonald et al. 2016), thus contributing to social change (Rojas-Pernia et al. 2020) and helping to build a society in which groups identified as excluded begin to function as its integral parts (Walmsley et al. 2018). One of the foremost benefits of inclusive research is that it has a significant impact on improving the quality of life of persons with intellectual disabilities (Nind and Vinha 2012; Walmsley et al. 2018).

Inclusive research also entails challenges. One should consider the risk of exploitation and manipulation of persons with intellectual disabilities in research, which might result from their vulnerability and powerless position in these particular circumstances (Juritzen et al. 2011; McDonald et al. 2017). Therefore, co-researchers without disabilities should accept responsibility for safeguarding the rights of participants and co-researchers with intellectual disabilities so that they may exercise autonomy and enjoy freedom from harm (Northway 2014). The fact that the research subject has little relevance for the lives of persons with such disabilities may also be a problem. Indeed, a premise which is often underscored in inclusive research is recognising that it should make a positive contribution to the image or life of persons with intellectual disabilities (cf. Nind and Vinha 2012; McDonald et al. 2016).

### 1.3. Different Approaches to Inclusive Research

With regard to the organisation of inclusive research, it is a crucial requirement to ensure that the research, as well as the tools and the material used, match the capabilities of persons with intellectual disabilities (Nind and Vinha 2012). Finding ways to empower vulnerable populations (Moll et al. 2020) and put persons with intellectual disabilities in control (Walmsley 2001) may be another major difficulty. It is underlined in the literature that the key principles and practice of co-design, as well as the extent of inclusion of people identified as disadvantaged (Moll et al. 2020), still lack clarity. Over the years, several approaches to inclusive research have developed (see Swain 1995; Bigby et al. 2014). Bigby et al. (2014) distinguish three basic modalities: advisory, leading and controlling, and collaborative group, which are characterised by varying degrees of participation of persons with disabilities, including intellectual disabilities. The collaborative group approach offers the highest level of participation.

### 1.4. Organisational Requirements and Strategies for Inclusion

To be truly inclusive, research should meet certain criteria. Walmsley and Johnson (2003, p. 64) suggest the following five principles: (1) the research problem, although it may be initiated by co-researchers without disabilities, should be formulated by co-researchers with disabilities; (2) the research should further the interests of co-researchers with intellectual disabilities; (3) people with intellectual disabilities should be involved in the research process and (4) have some control over it; and (5) the research questions, research process, and research reports should be accessible to people with intellectual disabilities.

The diverse strategies which facilitate thorough inclusion of co-researchers with disabilities include nurturing positive relationships (even before a person enrols in the project), acknowledging that they are experts by experience, allowing them to work at their own pace or use alternative, creative methods and teaching resources (e.g., photographs, illustrative materials, and recordings) (Moll et al. 2020). There are also a number of specific ways that may help deliver the successive stages of a research project. For instance, visual representation of data, group analysis (using focus groups in which co-researchers with disabilities participate), and familiarity with data collection are recommended in the process of analysing and interpreting collected research data (Kramer et al. 2011).

Although reports and studies describing inclusive research involving co-researchers with intellectual disabilities can indeed be found in the literature, few studies are concerned with non-English-speaking countries (not least because such research is not as widespread there) (see Rix et al. 2020) and allows for the direct viewpoints of co-researchers without disabilities as they reflect on the research process in its different phases. This report aims to partially address this gap.

## 2. Methodology

### 2.1. Study Design

The aim of this paper is to analyse the co-researchers' without disabilities experience of designing and conducting inclusive research, which involved persons with intellectual disabilities. The first two authors were involved in the study that was carried out as part of the project "Nothing about us without us. Inclusive research involving persons with intellectual disabilities", whose aim was to explore the experiences of people with intellectual disabilities associated with their self-determination. The project was conducted in cooperation with trained co-researchers with intellectual disabilities. One of the principal elements in the research process was the application and verification of the methodological and ethical premises of the participatory method in research concerned with intellectual disabilities. During the whole project, the first two authors (co-researchers without disabilities) prepared field notes and kept observation diaries to facilitate the process of this verification. As such, we wanted to reflect on our experiences regarding the training for prospective co-researchers with intellectual disabilities and inclusive research conducted in collaboration with trained co-researchers with intellectual disabilities, focusing on ethical and methodological considerations. The involvement of persons with intellectual disabilities in the conduct of the research made it possible to determine analytical categories formulated by the persons with whom the study was concerned and to preserve the meanings that research participants attributed to their views, actions, and experiences (Charmaz 2006).

### 2.2. Procedure

Collaboration with persons with intellectual disabilities was undertaken in February 2022 in two groups, composed in part of self-advocates who were affiliated with the Polish Association for People with Intellectual Disability (PSONI). Prospective co-researchers with intellectual disabilities were recruited through institutions dedicated to persons with intellectual disabilities, i.e., occupational therapy workshops and vocational activity centres, both of which come under PSONI. Initially, three meetings were held at PSONI facilities located in two Polish cities to introduce the aim and the objectives of cooperation. Out of the three approaches to inclusive research identified by Bigby et al. (2014), the researchers in the

current study were all involved in collaborative groups. Those interested in participating consented in writing and agreed on the rules of mutual communication as well as on the frequency, location, and time of meetings. As the first element of collaboration, researchers without disabilities delivered training (eight meetings spanning 17 h in total). Its scope was not limited to research issues, but, having a broader dimension, it served to enhance the social skills of the co-researchers so as to boost their functioning in different roles, including self-advocacy (Parchomiuk et al. 2023). The training was followed by two focus interviews to explore the experiences of prospective co-researchers with intellectual disabilities (Żyta et al. 2023). Those who were still interested in working together joined two research teams that operated in two cities in Poland but remained in close contact with each other. The teams jointly established the objective of the first project, centred around the self-determination of persons with intellectual disabilities.

The whole process of training and conducting the inclusive research was not free of numerous challenges related mainly to organisation (e.g., providing space and choosing time and dates of meetings), communication (among persons with disabilities, professionals working in institutions and and co-researchers without disabilities), and ethics (obtaining informed consent from participants and co-researchers with intellectual disabilities). At the time of writing this article, the project had been completed. Co-researchers with and without intellectual disabilities conducted and analysed three focus group interviews and 13 individual interviews with persons with intellectual disabilities. The results, communicated in an easy-to-read text, were published and presented at a conference involving persons with disabilities. The teams of co-researchers embedded in two Polish cities have already decided on two further research topics.

*2.3. Data Analysis*

Data were examined and interpreted using Braun and Clarke's six-phase approach to qualitative thematic content analysis to examine researchers' experiences related to the design and implementation of inclusive research (Braun and Clarke 2006). The information was based on field notes and observation diaries kept by co-researchers without disabilities (the first two authors). To provide a more holistic view of the collaborative experience, further research will include the perspective of co-researchers with disabilities.

The first phase of the analysis consisted of familiarising oneself with data, reading data, and noting down initial ideas. In the second phase, initial codes were generated, whereby interesting features of the data were coded systematically across the entire data set, and data relevant to each code were collated. The codes were discussed by the authors, who suggested multiple coding if there was no consensus. In the third phase, topics and subtopics were proposed to group the codes; codes were collated into potential themes, while all relevant data were gathered under each potential theme. In the fourth phase, a step backwards was made once the topics and subtopics had been obtained so as to review the codes or even extracts of information to refine them and ensure that they were consistent with the data, whereby necessary adjustments were made. Here, the intention was to verify whether the themes were valid in relation to the coded extracts and the entire data set, generating a thematic map. In the fifth phase, the themes were defined and named. Ongoing analysis was carried out to refine the particulars of each theme and the overall story emerging from the analysis in order to arrive at clear definitions and names for each theme. Finally, a report was produced. Subsequently, we arrived at a clear and concise determination of the main themes and their respective constituent themes, the role each one plays, and a set of literal examples that helped us to describe and understand the themes.

**3. Findings**

The analyses identified four superordinate themes (building relationships in the research team; opportunities and constraints associated with the implementation of projects; institutional barriers; and the importance of the role of a co-researcher without intellectual disabilities), each followed by several constituent themes (Table 1).

**Table 1.** Superordinate and constituent themes.

| Building relationships in the research team |
| :---: |
| Building trust and gradual integration of the team<br>Conflicts between co-researchers with intellectual disabilities<br>Setting boundaries |
| Between (1) challenges and (2) potential of co-researchers with intellectual disabilities: opportunities and constraints associated with the implementation of projects |
| Constraints:<br>- Discouragement;<br>- Difficulties in remembering, reading, and reacting flexibly to the answers from the respondents<br>Opportunities:<br>- The knowledge of researchers with intellectual disabilities about the community of persons with disabilities;<br>- Bridging the gap in research involving persons with intellectual disabilities;<br>- Developing competencies;<br>- Discovering one's capabilities and using previously acquired knowledge;<br>- Nurturing responsibility;<br>- Empowerment |
| Institutional barriers relevant to the participation of persons with intellectual disabilities and the quality of cooperation |
| Selection of participants for the co-researcher team<br>Limitations of access to certain potential research subjects<br>Interference by institutional staff in training and inclusive research design |
| The importance of the role of co-researcher without intellectual disabilities |
| Tailoring tasks to the abilities and needs of persons with intellectual disabilities<br>Sustaining motivation in researchers with intellectual disabilities<br>Negotiating one's own role in an equal/unequal relationship with co-researchers with intellectual disabilities—resolving the question of the limits of interference<br>Doubts concerning the inclusive nature of research<br>Enhancing one's own competency and commitment |

Source: own research.

### 3.1. Building Relationships in the Research Team

The first superordinate theme concerned the relationships within the co-researcher team. In this respect, an analysis of the collected material yielded three constituent themes. One of them is the issue of gradual trust building and team integration. The process was initiated already during the training course when the participants (who had no previous opportunity to get to know each other better) started working together. Undoubtedly, the word "cooperation" gained increasing significance as persons with intellectual disabilities became familiar with one another and improved their skills. As early as the training phase, participants would collaborate, help, and take mutual interest in each other:

> *Magda*[1] *practices writing in Word. She quickly developed the ability to write Polish characters. It is difficult to tear her away from this activity. She is instructed by Anna and a little by Karol. We show Magda how to enlarge the font, change its format, insert a chart, a table (Anna remembers how to insert a table). (Training #3)*

> Anna sees *Magda's difficult behaviour and tries to correct it by bringing it to her attention. (Training #4)*

The second constituent theme revolved around tensions and conflicts between co-researchers with intellectual disabilities. The relationship dynamics within the research teams did not always follow a linear path towards full inclusion and collaboration. Tensions and conflicts with varying degrees of severity arose between persons with intellectual disabilities at different stages of the work (especially during training):

> *Carolina was absent today. Other participants commented openly on her absence, stating that it was good that she was absent, that she was too conflict-prone and insincere*

*(running others down), and that it would be better if she did not come to project meetings anymore. (Training #4)*

Conflicts made it necessary to nurture positive, symmetrical relationships. It was vital to conduct activities without competitive elements to satisfy the need to be appreciated and to promote co-responsibility for the research.

Another constituent theme generated in the relationship analysis concerned the demarcation of boundaries between co-researchers with disabilities and co-researchers without disabilities. One note reads:

*Marek became very involved in conducting interviews. In the course of last week, he invited three of his colleagues to participate in the research, and we conducted the interviews together, each time in the late afternoon after work. He calls frequently and asks when we could make an appointment with another person. Today, he called after 9 p.m. I think he feels a little offended when I suggest we talk about it at our regular meeting. (Project Meeting #6)*

In the situation described, the co-researcher without intellectual disabilities was presented with a dilemma: to benefit from Marek's commitment and devote more time to the project (at the expense of time for other commitments, family life, etc.) or to set boundaries and define a reasonable time for project collaboration. In these circumstances, there was no obvious solution. Marek seemed to lack people close to him with whom he could spend his free time. The research group met that need to some extent, but he expected more involvement from others, not only with regard to research tasks but also in terms of informal relationships or friendships. It was necessary to establish certain general rules within the group (e.g., arranging interviews a week in advance, taking into account the preferences of each co-researcher and interviewee, as well as the times we could contact each other by phone).

*3.2. Between Challenges and Potential of Co-Researchers with Intellectual Disabilities*

The second of the generated superordinate themes focused on persons with intellectual disabilities. It comprised two constituent themes: (1) constraints and (2) opportunities associated with the implementation of projects. First, we learned about the challenges faced by persons with intellectual disabilities both during training and while the research project was in progress. Such issues had multiple backgrounds, including learning challenges (in terms of memory, attention, perception, and abstract ideation) and emotional issues on the one hand, and the closely linked obstacles in their surroundings, which resulted in the absence of various life experiences (such as using a telephone or computer).

*Karol got tired because he had a lot of problems understanding. It was apparent that he was unable to understand longer verbal communication (approx. 3–4 sentences). When he did not understand, he'd switch off. (Training #2)*

*Anna is very nervous and reacts . . . often swearing to noises from the outside. (Training #4)*

*It turns out that both [Magda and Anna] are unable to use a traditional watch. They cannot tell the time. (Project Meeting #4)*

The second, more extensive area relating to the implementation of inclusive research involving co-researchers with intellectual disabilities underscored their ability to participate in the project. The categories identified here included the knowledge of the researchers with intellectual disabilities about the community of persons with disabilities and bridging the gap in research involving persons with intellectual disabilities. In this respect, the potential of persons with intellectual disabilities was high and enabled a better design and implementation of the project:

*Anna helps the most [with selecting potential interviewees]: she has much knowledge about people from OTWs, and by interacting with them, she obtains a picture of both their character and their skills as well. (Project Meeting #1)*

Our experience has shown that persons with intellectual disabilities demonstrated their capabilities from the very first training sessions.

*[Co-researchers with intellectual disabilities] stepped into the role of researchers brilliantly. I didn't expect that at the first meeting. Everyone found a person at the VDC [Vocational Development Facility] they conducted a mini-interview with and were able to talk about it. (Training #2)*

Their skills increased with the training and subsequent stages of inclusive research; for instance, most learned how to use email and a voice recorder, and they knew what rules should be followed when conducting and analysing interviews:

*We listen to the recorded interviews, looking for errors. Almost all of them are identified by Magda and Anna. When, in one of the interviews, the researcher forces the respondent to answer, citing his consent, Karol points out that since the respondent has given his consent, he must answer. Magda immediately notes that the researcher's behaviour is inappropriate. (Training #7)*

*Magda demonstrates a remarkable ethical stance. When Karol's and my comments go beyond acceptable norms, Magda responds by making a remark to Karol. I also acknowledge my mistake and emphasize Magda's positive approach. In the course of the collaborative process, I continually noticed Magda's progress related to her participation in the project. (Project Meeting #2)*

*In the beginning, Anna had a problem switching from external, critical evaluation, which was required when analysing the transcription, to establishing what the respondents said, but she's gradually learning to do so. (Project Meeting #5)*

Systematically, as the training went on, we also observed growing intrinsic motivation for work among persons with intellectual disabilities, which concretised itself in increased activity ("Their activity in the training sessions steadily increased as they continued", Training #1) and self-sufficiency ("Aleksandra, who previously always avoided speaking in front of others, today volunteered to conduct the interview herself", Training #2).

Over time, the co-researchers also showed an increasing sense of responsibility for their behaviour and the performance of the team. We believe that this was one of the key achievements of the training and the project.

*It is a testament to their responsibility and commitment that they managed to do the interviews after the first meeting, where homework—such as conducting interviews—was only preliminarily mentioned. (Training #1)*

The responsibility and growing maturity of the co-researchers with intellectual disabilities was evident in their approach to the meetings and tasks. Gradually, we reached a point in our collaboration when one of the participants became responsible for arranging a meeting and conveying important information.

*Anna is already waiting for me and has been informed that the session is being held in another room because the other one is occupied. She organizes the others. She also informed me that some people have gone to see one of the participants (who took part in the group interview) who does not want to attend the workshops to talk to her. (Project Meeting #4)*

Towards the end of the training, one could see that the co-researchers with intellectual disabilities had grown into the role they were to perform. They had ceased to be randomly selected persons who were faced with a task which, being completely unfamiliar, was a challenging one. They understood (perhaps largely intuitively) what research was and what their role would be in the entire undertaking. They gained proficiency in referring to their life knowledge. It turned out that such knowledge and interests may prove important for one's tasks as a researcher:

*We brainstormed the question: what do our interview questions need to be? Everyone was able to contribute something on their own. Magda, known for her medical interests,*

*highlighted sensitive issues, such as asking about illnesses, cancer, and medication taken. Karol elaborated on the issue of the researcher's sensitivity (he used the phrase "they must be alert") to the emotions of the respondents. With my guidance, they considered people who cannot speak or cannot hear. (Training #6)*

As the sense of responsibility on the part of co-researchers with intellectual disabilities grew, the role of the head of the institution in arranging project meetings became superfluous. When co-researchers with intellectual disabilities happened to be committed elsewhere, they would comprehensively justify their absence or lateness. They were upset about events which made it difficult for them to attend the meetings:

*At the beginning only Anna is present, Karol and Magda have gone shopping. We review what we did before (Anna says she remembers). Magda and Karol, both late, come into the room, apologizing for not being on time. They're clearly agitated because one of their friends delayed them. Magda states that she needs to report this to the therapist. Karol says that he won't get involved. (Project Meeting #6)*

On the one hand, participation in the training and the project offered an opportunity for the co-researchers to develop their potential and discover it, while on the other, it enabled them to apply the expertise deriving from other forms of social activism (e.g., self-advocacy):

*Anna has ideas for research with a broader dimension. You can see her experience in social activities (she's currently at the radio). Her only problem is formulating thoughts. When I ask about something, she considers it very thoroughly and usually suggests the right answer with a movement of her head. [...] Interestingly, that intuition works quite well. (Training #5)*

*Karol and Magda step into their roles better and better, and they're more open. Karol has knowledge from different areas of life (it is not structured). It is apparent that issues related to studying (university work) are familiar to him. This is evident in the terms used (professor, defence, BA thesis). Anna recalls the skills they once received training in (perhaps as part of self-advocacy). Magda has a rich vocabulary and broad interests. (Training 3)*

The knowledge and the skills of the co-researchers with intellectual disabilities would occasionally surprise us as researchers:

*I don't know where she [Magda] got such knowledge from, e.g., one of the research questions [which she has formulated] was about the concerns in one's first pregnancy and fear of abortion due to defects in the baby. There was also a question about the use of oral surgery, the procedure and the subsequent symptoms. (Training #6—research design)*

*Karol knows words such as "prelegent [lecturer/speaker]". I am surprised at his perceptiveness when I cannot decipher an abbreviation while he associates it with the full designation provided previously. (Training #8)*

Another category related to the challenges and opportunities of inclusive research concerned the empowerment of the co-researchers with disabilities. Throughout the course of the training, the research tasks became less challenging for the co-researchers with intellectual disabilities. It was crucial still to divide the research tasks relative to the individual traits of the co-researchers with intellectual disabilities, which, suitably situated in subsequent research work, became a resource (e.g., the reticence of the researcher affords the respondents space to speak). We began to realise that each person with intellectual disabilities would not respond to the same set of tasks, and instead, we opted for more flexible study paradigms.

*Anna and Karol are very animated and interested, you can see the positive changes over time. Karol is keen to remember what needs to be done at home, and this does not need to be written down. He really enjoys transcribing in Word. (Training #4)*

*I didn't think that [the trainees] would engage that much in the exercises involving other people. It turned out that conducting "mini-interviews" with their colleagues from the Vocational Development Centre was an opportunity to test oneself in a new role—that of a researcher and to interact with someone of their own choosing. Although the topic was imposed (they were to find out about the interviewees' previous experience of participating in scientific research), it seems that the ability to choose who to interview and to act as an expert was important for them. (Training #2)*

### 3.3. Institutional Barriers in the Process of Research

Another superordinate theme concerned institutional barriers relevant to the participation of persons with intellectual disabilities and the quality of cooperation, in which three categories were identified. Selecting participants for the co-researcher team was the first of those. In the project discussed here, the initiative to carry out the training and the project came from the researchers without disabilities. Having obtained permission from the institutions working with persons with intellectual disabilities, we—as university researchers—had to rely on the decisions of the competent officers and therapists at those institutions regarding persons who would be asked to participate in the project:

*It's yet another meeting [with the representatives of the institutions] about the planned training of potential co-researchers with disabilities. We [university researchers] provided an estimated number of participants in the training and described what our work would involve. We were given information about the potential participants, whereby emphasis was placed on their skills (which were higher compared with other participants). We argued that we were not interested in having only such [more competent] persons on board.*

Ultimately, the persons to join the team of co-researchers were selected largely by the representatives of the institutions. They did so based on their own criteria, probably considering the cognitive and social skills of the prospective participants. The selection was thus dictated in the sense that both researchers without disabilities and those with intellectual disabilities had to comply. Such a practice is self-evident within the institutional framework. Admittedly, persons with intellectual disabilities voluntarily consented to participate in the training activities and, theoretically, their consent was an informed one. However, one may ask about the extent to which persons with intellectual disabilities who joined the training were aware of the expectations and the tasks ahead, having never participated in such an undertaking. A number of persons with intellectual disabilities have no opportunity to gain experience in independent decision making. These obstacles are not only due to the challenges of having an intellectual disability but also to the environmental attitude towards the independence and autonomy of that group. The socialisation within institutional living to which persons with intellectual disabilities are exposed most often leads to dependence and lacks of a sense of agency, which impairs proactive strategies for coping with the challenges of daily life and compels them to adopt a passive approach. They have to confront entrenched notions, which determine their life situation and which can prevent them from becoming active as individuals.

Limitations of access for potential respondents made another constituent theme. It was to our advantage that, having completed the training phase, we were already more embedded in the institutions as researchers and were familiar not only with the trainees themselves but also with other charges of the institutions; in addition, we were able to rely on the knowledge of the co-researchers with intellectual disabilities about potential research subjects. Nevertheless, the approval of the staff member responsible for cooperation was required whenever a person was invited to be interviewed. Although this was not a major obstacle in reaching potential research participants, we felt that we were not completely at liberty in this respect.

Another constituent theme relating to institutional barriers concerned the interference of institutional staff in training and inclusive research design:

*At the outset, I received information from the Head of the institution that participants get nervous before meetings. She suggests using biscuits and reducing the requirements (?). It turns out that there is a problem with Magda, who clearly does not want to participate. When I suggest that she withdraw, the Head vehemently denies it, claiming that they should get accustomed to duties. The problem is that so far, no one has introduced Magda into dealing with duties, and everyone is filling in for her. (Training #4)*

In this sense, we were doubtful about who actually made decisions regarding the participation of persons with intellectual disabilities in the project. Although we applied the principle of double consent (the consent of the institution directors was a prerequisite, but it was neither a sufficient condition nor was it the same as the consent of the persons with intellectual disabilities themselves), we were uncertain whether the decision taken by the persons with intellectual disabilities did not result from an asymmetrical, hierarchical relationship or involve any form of obligation or coercion or the use of a privileged position by the staff. Because persons with intellectual disabilities are used to complying with the decisions of the personnel employed in institutions, we took particular care to convey that the activities proposed as part of the research were not compulsory and that they could opt out. For some persons, being invited to join the team was initially a task they were expected to complete because such was the decision of the institution's head or a parent. Perhaps this should be seen in positive terms as promoting dutifulness or counteracting passivity. However, it is difficult for us to assess how much being obligated in this fashion by a superior contributed to enhancing one's sense of responsibility and how much this was due to the growing satisfaction after successive team meetings.

During our meetings, the institutions which the co-researchers attended did not display any particular interest in the project. Usually, we would not be faced with obstacles or negative attitudes (apart from the occasional reluctance to carry out project-related tasks for organisational reasons, e.g., lack of a suitable room), but we also received no support or clear signals from the staff, which would confirm that our endeavours were meaningful and useful. When an easy-to-read text was produced as a measurable outcome of our undertaking, there was no interest in posting it on the websites of the involved associations or institutions. Perhaps assisting other participants with homework may be seen as a form of supporting them in what they were doing. Still, in some cases, the participants were not supported in their work, but instead, the tasks were completed for them, which demonstrates that the staff failed to acknowledge the capabilities of persons with intellectual disabilities and implies that a medical, care-providing model of disability still applies.

*It turned out that homework was not independent. It was done with extensive help at the OTW. (Training #2)*

One can hardly speak of cooperation with the staff, but over time, they noticeably expressed appreciation of our activities, though only verbally. Surprisingly enough, the staff failed to recognize that our activities could be integrated into the institutional system aimed at the social rehabilitation of persons with intellectual disabilities. It appears that our research proceeded on the sidelines of the main current activities, sometimes even interfering with them.

### 3.4. The Role of Co-Researcher without Intellectual Disabilities

The last of the identified superordinate themes concerned the importance of the role of a co-researcher without intellectual disabilities. Within this area, four constituent themes were singled out. The first was flexibility and adaptation of tasks to the abilities and needs of persons with intellectual disabilities. To avoid information overload, we divided the training into separate modules (e.g., data collection and data analysis). In order to minimize engaging with abstract material, the training sessions were designed to enable persons with intellectual disabilities to work on transcribed interviews and prioritised conversations on topics related to the researchers' knowledge, everyday events in which they partook, or specific interviews that they had conducted. When working with persons with intellectual

disabilities, it was vital to remain sensitive to their needs, to recognize signs of a low sense of agency, or to respond to behaviours that might lead to withdrawing from the project:

> *He is very active, although as the session goes on when I speak, he switches off slowly and resumes his stereotypical activity (plucking at his trousers). He would like very much to talk about his experiences, not necessarily related to the subject. Once again, I find out that the exercises need to be as practical as possible. This is important for the future editions of the training. (Training #8)*

Another constituent theme centred around negotiating one's own role as a co-researcher without disabilities. The role of the researcher without intellectual disabilities changed in the subsequent stages of collaboration. The training stage was dominated by the non-disabled researcher, who decided on the topic of instruction, selected the methods and the means to deliver it, set specific tasks, and expected them to be completed. The essential premises and training modules had been designed even before the co-researchers with disabilities were selected. Once the participants had been enrolled and the training started, it became apparent that flexibility and modifications were needed at various points in view of the participants' skills and their responses to particular tasks. Openness to modification was pivotal for the subsequent tasks, as it helped to build mutual understanding, reduce the anxiety of persons with intellectual disabilities, and reinforce a sense of empowerment in the persons involved. Modifications were introduced by persons without disabilities and included extended time to cover a particular module (e.g., disseminating results, attending conferences, and conducting mock interviews), either due to a high interest in the module or the difficulties experienced, replacing tasks with others (e.g., more specific ones), dispensing with homework that might be difficult, or abandoning a task if its completion—in the opinion of the participants—was beyond their capabilities and prompted discouragement. Already during the training stage, the co-researchers with intellectual disabilities would gradually recognise that they had distinct capabilities and could contribute differently to prospective research. Also, they began to influence the content of the training themselves, as illustrated in the following note:

> *Today, we continue to practice digital skills. Jan [the oldest participant, aged 64] seems disgruntled from the outset. He is a leader in the self-advocacy group, but this is the area in which he feels least capable. While the others are engrossed in texting each other, setting up email accounts and using voice recorders on their mobile phones, he can't keep up. He has an early-type phone—it's not a smartphone, it doesn't have internet access, and it's difficult to write texts on it. He uses other people's devices, but he's clearly annoyed. He eventually refuses to be helped with texting and reading messages and asks to be called when needed. He will happily work with text (being a skilled reader) but wants it printed out for him. We accept this request. We affirm that it is normal that everyone will use different resources. (Training #5)*

As authors of the training proposal, we made efforts to involve the participants and to learn about their potential and needs in order to respond in the most adequate fashion. However—as articulated in our notes—we feel that we did not provide a sufficient number of activities, which would evoke positive emotional responses, which relied on specific experiences (such as the interview simulation), incorporated elements of humour, or provided an opportunity to relieve the tension that may have initially accompanied persons with intellectual disabilities. Another issue was taking continual care—especially while conducting the research—to ensure that the voices of persons with intellectual disabilities, who had by then become co-researchers, counted as thoroughly valid. That influence was evident at various stages, but an eloquent situation during training evinced something contrary to the desire of persons with intellectual disabilities for decision making. It may be interpreted twofold: as a manifestation of subordination to a superior, but also as a learned life attitude which assumes that non-disabled persons know better and, therefore, have power:

> *The researcher (myself) was conducting the research (training role plays) while Anna was the respondent. The situation showed all possible violations of ethical principles, e.g., that consent was only given by the, staff but the person with disabilities was not asked. "The researcher" assured the "subject" that they should fill in the questionnaire because the consent was given by the institution. Following the role-play, questions were asked about the inappropriate behaviour of the "researcher". It was decisive for Anna that consent had been given by the institution, and she stated that it had been given she should complete the questionnaire. In my opinion, this is another manifestation of subordination to non-disabled people. What the manager says is conclusive, even if it provokes internal objections. (Training #2)*

When we listened to the stories from the lives of researchers with intellectual disabilities, we had the impression that they had been poorly equipped to claim their independence and their capacity to decide and take advantage of "shared decision-making" on an equal footing with co-researchers without disabilities. Persons in the most immediate surroundings of persons with intellectual disabilities tend not to believe that they are sufficiently competent, which is why they feel entitled to make decisions on their behalf, "for the good" but not always "with the consent" of those concerned. This ultimately leads to an absence of initiative, submissiveness, or an internalised image of oneself as a "recipient" of support. This typical socialisation pattern within family and institutional environments also resonated in our project:

> *In Magda's case, there's tremendous dependence on her parents' will. She has a parental lock put on her computer and is afraid to open an email account. When medical examinations are discussed, she states that it is the parents who take care of everything and decide about everything. She never gets to consent (nobody asks her). (Training #3)*

Creating a research team involving persons with intellectual disabilities compelled us to rethink the role of our own ability and the resulting unequal relationship. Situations where power relations changed were an interesting experience. From time to time, we would be confronted with the limitations of what is seemingly a privileged social position; for instance, our status in research into intellectual disability was questioned. On more than one occasion, it was thanks to the researchers with intellectual disabilities that we gained visibility and trust in certain communities (which was important in recruiting research participants).

The misgivings on our part related to the above as to whether the research was genuinely inclusive and became one of the themes identified in the course of analysis:

> *We started the meeting with a brief reference to a recent interview. Since Karolina was additionally participating today, we drew her attention to the fact that she should remember not to judge the respondent's statements during the interview. I had the impression that Karolina did not consider it an element of good practice in general but as a task she has to do because she is told to. I am not sure that the conducted research really affords a sense of agency to everyone. Sometimes, I feel that for some, it is a chore or an opportunity to escape doing daily tasks at the Workshop. (Project Meeting #5)*

The final constituent theme focused on the opportunities to enhance the potential and commitment of researchers without disabilities. For us, collaboration with persons with intellectual disabilities as researchers was an opportunity to boost our own research skills. In the first place, this means acquiring more extensive knowledge of how persons diagnosed with intellectual disabilities tackle tasks that were previously unavailable to them or to which they were not invited. We found that the researcher has to be highly sensitive to indirect communication, seemingly unrelated to the issue at hand. What may appear to deviate from the topic is sometimes a substitute (or "circular") attempt at talking about it. Our assessment of the statements made by persons with intellectual disabilities can often be too precipitous and thus fail to interpret them in a broader context:

> *Today, we were analysing a group interview. I [co-researcher without disabilities] read excerpts from the transcript, and the co-researchers with disabilities commented on the*

*statements. Sometimes, those were simple recapitulations, but more often than not, they went much further than I would have thought on my own. They inquired and looked for reasons, citing their own experience, the experience of a particular respondent or the general situation of people with disabilities, who are often dependent on others: families or therapists. Sometimes, they questioned the answer, suggesting that someone might be ashamed of certain behaviours or their incapacity to decide for themselves. (Project Meeting #8)*

*Her [Magda's] statements repeatedly feature comments such as "they're afraid", it's about leaving the family home, living somewhere else, or starting a family. (Project Meeting #5)*

*I'm glad that we are finally starting the actual project [after the training]. In today's session, we selected our topic. [. . .] I'm not alone in this process, but rather encouraged [by co-researchers with disabilities] and empowered that there are others who feel the same way. (Project Meeting #2)*

We are convinced that the participation of persons with intellectual disabilities as co-researchers enabled more accurate interpretations of the material we obtained during the interviews. We have learned to hear that voice, appreciate the sensitivity, and take the non-obviousness of interpretations into account. Not unlikely, what we would have previously attributed to want of attention or interest may have been an attempt to satisfy the need to be heard, to use the voice that one was granted. Persons with intellectual disabilities are more often required to be listeners. They seldom have the ability to share their opinions and thoughts, and even more rarely are they invited as experts (even on their own affairs). This research collaboration provided them with such an opportunity, which entailed sharing knowledge that is important to them, issues which have been essential in their lives, and which make up their life experience. However, the benefits do not end there, as the collaboration was also "an empowering journey" for the co-researchers without disabilities. We came to feel important in the process because, by listening to the voices of persons with disabilities, we became convinced that we could truly make a difference together.

## 4. Discussion

### 4.1. Collaboration in Inclusive Research

The analysis of the collected material demonstrated the major importance of collaborating with co-researchers with intellectual disabilities at different stages of research from planning to analysis and dissemination. It is clear that co-production is a relational process, and as such, special emphasis should be given to ensuring that the process does not perpetuate inequities.

In the current project, substantial attention was paid to building positive relations between co-researchers. We tried to be responsive to the needs of co-researchers with intellectual disabilities and concentrate on their strengths, not deficits. This is an important aspect of all-inclusive studies involving persons with intellectual disabilities (Townson et al. 2004; Kramer et al. 2011), but, based on the findings of this study, it is not always easy and straightforward. It is important to be able to set boundaries to avoid situations in which members of a team of co-researchers might feel personally neglected. On the other hand, creating occasions to build an integrated and cohesive team is of crucial significance. The findings indicate that both co-researchers with and without disabilities value not only their professional roles as researchers but also making new friends, and informal contacts that can be built. Collaboration between co-researchers has begun to transcend the formal dimension as we have become friendly. Our meetings started with discussing life issues, which we considered important. We know one another quite well, as much as our joint willingness to share experiences allows. This is no obstacle to project-related activities. Teams of non-disabled researchers also see progressive integration, which is not unique to our experience (Nind and Vinha 2012; Riches et al. 2017).

### 4.2. Benefits of Inclusive Research

We have observed that the project had positive significance for the co-researchers with disabilities, the co-researchers without disabilities, and the quality of inclusive research.

The project tasks carried out by co-researchers with intellectual disabilities had an enriching effect on the latter. Some assumed multiple roles, which enabled them to show their potential (e.g., acting in a theatre), though in certain instances, the scope of their tasks was limited. In either case, participation in the project expanded their repertory of skills (e.g., communicative, digital, and social) and contributed to empowerment. Meta-analytical studies and investigations addressing the experience of inclusive research confirm that participation in such undertakings creates opportunities to enhance the potential of persons with intellectual disabilities (Dorozenko et al. 2016; McDonald and Stack 2016; Puyalto et al. 2016; Frankena et al. 2018; Armstrong et al. 2019; Hewitt et al. 2023).

Co-researchers without disabilities were also subject to a development process. One of the important objectives of our analyses was to determine the role researchers without disabilities play in ensuring that co-researchers with disabilities have a genuine impact on the study and at each stage of its activities. Our findings indicate that the participants with intellectual disabilities conformed to the decisions of others and were overall non-critical and dependent. In therapy and education alike, their refusal or disapproval is approached as a manifestation of disturbed behaviour rather than agency. The arbitrariness of intellectual disability and the medicalisation of behaviours are tools of social control and power relations. The question which arises in this context concerns the possibility of changing the social standing of persons with intellectual disabilities. It is construed as a vital element in the added value of such research (Walmsley et al. 2018; Beighton et al. 2019; Georgiadou et al. 2020), but it seems that the expectations involved do not always correspond to realities, at least in Polish circumstances.

The reach of inclusive research is still considerably limited, whether in educational and rehabilitation environments or in the scientific milieu. In the latter, they are regarded as a kind of experiment and a curiosity. In the former—as reported above—such research is no more than an ancillary activity, which hardly fits into the established institutional agendas. According to Nind (2016), it is possible for the paradigm of inclusive research to combine sound science with good practice. The status and significance of such research situate it at the intersection of practice and science. Buchanan and Walmsley (2006) note that such research must not become an exclusive undertaking that remains separated from the broader research community, approaches, and debates. Its integration into practice and science may, on the one hand, promote a change in societal attitudes towards disability, yet on the other, it requires a cultural change within various milieus, a shift oriented towards diminishing the primacy of experts and researchers (cf. DiLorito et al. 2018). Intensification of inclusive research in Poland—in view of its relevance for the policies that benefit persons with disabilities—would require large-scale financial support as well as the formation of future researchers and practitioners (students) who would conduct such studies (O'Brien et al. 2022).

We believe that the inclusive nature of the research makes it particularly valuable. The very choice of the topic and research problems was informed by the needs of persons with intellectual disabilities, which were not approached from an external (non-disabled researchers) but an internal perspective (researchers with disabilities, so persons who had experienced disability themselves and could more easily "step into the shoes" of the respondents) (cf. O'Brien et al. 2022). In this project, the co-researchers with intellectual disabilities turned out to be very creative in developing research topics. We wanted to determine what kind of research they wished to have performed and conducted discussions, which showed that the greatest challenge was barriers to their self-determination. This led to designing the research problems and the whole study. In their interpretations of the material, the co-researchers with intellectual disabilities drew on their own experiences and life attitudes, enriching the analytical process. Similarly to the co-productive research conducted by Armstrong et al. (2019), we see the added value of our research primarily

in the fact that each co-researcher was able to make full use of their knowledge, skills, and experience, while the research yielded a useful outcome (recommendations were developed). There is a certain risk associated with addressing difficult matters which can personally concern co-researchers with intellectual disabilities, as it might affect their well-being (Richardson, in Bigby et al. 2014). The authors recognised that the co-researchers in the project were capable of keeping their own experiences separate from the data collection process, which allowed them to remain objective and prioritise the experience of the respondents (Rojas-Pernia et al. 2020). However, it remains unquestionable that the discussion on the well-being and comfort of the co- researchers should have its due place within inclusive research (cf. García Iriarte et al. 2023).

*4.3. Overcoming Challenges*

Collaboration with co-researchers with intellectual disabilities has also entailed dilemmas. When analysing such projects, it is advisable to avoid over-generalisations that may lead to a kind of instrumentalisation of inclusive research and co-researchers themselves (Goodley, in Björnsdóttir and Svensdóttir 2008). Inclusive research represents a domain of social participation for persons with disabilities that is accessible to few, just as is the case with non-disabled researchers. Goodley and Moore (2000) drew attention to the entanglement of non-disabled researchers in a highly abstract discourse that focuses on noticing individual and collective development of co-researchers, which may be an attempt to seek justification for such research given the multiplicity of other traditional approaches. Such an entanglement carries the risk of departure from the core objectives of inclusive research, the determination of which are, after all, central to the collaboration between co-researchers with and without disabilities. At this point within this study, we became aware of the imbalance of the data being collected only from the co-researchers with intellectual disabilities. It was therefore crucial to create a space for personal and collective evaluation of engagement, which we created through joint reflection on participation in the project. This resulted in a paper (Chamera et al. 2023) developed in an accessible format and published in a major journal in the field.

Just as other researchers (e.g., Björnsdóttir and Svensdóttir 2008; Lloyd et al. 2019), we also found it challenging to empower co-researchers with intellectual disabilities to exercise control over the research process and the reporting of findings. To ensure that control, we sought to comply with the interests and choices of co-researchers with intellectual disabilities (regarding research problems, research participants, and research location). Publicising research findings in an easy-to-read format proved an important solution. The prospect of publishing a paper in a scientific journal was not attractive to the co-researchers with intellectual disabilities, but they were very interested in creating (as co-authors) a text for a magazine that is well known in the community of persons with intellectual disabilities. Indeed, they saw it as a major achievement, as the work of collecting the results and analysing them became more tangible, while the final publication of the text in the magazine (with photographs and bios of the authors) became a source of satisfaction and pride. This demonstrates that the benefits associated with inclusive research are not equivalent, whereas any comparisons of the kind miss the mark, even though they are attempted, especially with respect to remuneration and promotion (Nind and Vinha 2012; Strnadová et al. 2016).

*4.4. Limitations*

The study presented here is not free of limitations. It should be underlined that it concerns only one research project and a small, homogeneous group of co-researchers with and without intellectual disabilities. It would thus be worthwhile to undertake investigations with other co-researchers with and without disabilities, who would be diverse in terms of socio-demographic characteristics as well as experiences in conducting inclusive research (different approaches to inclusive research and varying length of time spent conducting such investigations). It would also be valuable to expand the study

groups to include co-researchers with intellectual disabilities as well as self-advocacy and other activist communities dedicated to persons with intellectual disabilities. Alternative methods of data collection, such as photovoice, would also be necessary to enable the participation of persons with severe and multiple disabilities (e.g., Wos and Baczała 2021; Rojas-Pernia and Haya-Salmón 2022).

*4.5. Implications*

As for practice, the study may serve other researchers who are currently engaged in research with persons with intellectual disabilities or are considering undertaking such studies. First of all, the endeavour is worthwhile, as it benefits not only the research itself (high degree of practical utility, significance for the community of people with disabilities, easier access to research subjects, and sometimes greater openness of the respondents thanks to the presence of co-researchers with intellectual disabilities) but also the co-researchers with disabilities (empowerment, mobilisation, activation, and development of competencies) and without disabilities (increased awareness and a sense of purpose). Secondly, the project demonstrates that training co-researchers with intellectual disabilities is an important step in the process of inclusive research. Not only does it integrate the group of prospective co-researchers, but it also enables them to discover their strengths or equips them with the necessary skills to conduct research. Thirdly, this study makes it possible to formulate guidelines for the inclusion of persons with intellectual disabilities in research. In our opinion, it is vital that the researcher without disabilities continually strives to ensure that persons with intellectual disabilities may exercise control in ongoing projects, from identifying research problems (brainstorming, discussion, and voting among members of the research team proved productive in our project), through the research process itself (taking care that co-researchers with disabilities feel empowered to decide who will participate in the research and what questions will be asked) to reporting research results (ensuring that the reported findings are accessible to persons with intellectual disabilities). Accessible publications both in high-impact journals as well as communicating the results at a conference, on the institution's website, or in a popular scientific magazine are equally indicative of achievement for co-researchers both with and without intellectual disabilities.

**Author Contributions:** Conceptualization, K.Ć., M.P. and A.W.; methodology, K.Ć., M.P. and A.W.; validation, K.Ć., M.P. and A.W.; analysis, K.Ć., M.P. and A.W.; investigation, K.Ć. and M.P.; writing—original draft preparation, K.Ć., M.P. and A.W.; writing—review and editing K.Ć., M.P. and A.W. All authors have read and agreed to the published version of the manuscript.

**Funding:** This research received no external funding.

**Institutional Review Board Statement:** The study was conducted in accordance with the Declaration of Helsinki, and approved by the Research Ethics Committee of the University of Warmia and Mazury in Olsztyn (postal code 4/2022; date of approval 14 March 2022).

**Informed Consent Statement:** Informed consent was obtained from all subjects involved in the study.

**Data Availability Statement:** Data are contained within the article.

**Conflicts of Interest:** The authors declare no conflicts of interest.

## Note

[1] For the purposes of this paper, the names of persons involved in the research have been changed.

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
