# Peer review of "Cooperation with Persons with Intellectual Disabilities: Reflections of Co-Researchers Associated with Conducting Inclusive Research"

_socsci, doi:10.3390/socsci13030136_

Round 1

Reviewer 1 Report

Comments and Suggestions for Authors

This is a very original, important study which provides readers and funders/policy-makers with valuable findings on the benefits and challenges of working with persons with cognitive disabilities as co-researchers. There are aspects in the methods and discussion section that should be improved in order to strengthen the clarity and persuasiveness of findings and arguments.

Some detailed suggestions for clarification and elaboration are:

Abstract

·         Last sentence is too long, should be split in two.

Intro

·         L. 25: suggestion for further elaboration: in which respects (if any) are there different challenges between involving persons with intellectual disability and others, e.g. with patients with physical diseases? is there literature on children co-researchers that might also give some insights that could be useful?

Method

·         L. 103: provide more information on this project in the beginning/study design section would make sense in my view: when was the project conducted; what was the main aim of the project?

·         L. 107: specify: which premises?

·         L. 136: how many interviews, how long did they last, where they transcribed? eventually, can you characterise the persons with intellectual disability a bit more (age, degree of disability, etc.)?

·         L. 145 “the teams”: specify

·         Overall, the methods section should be checked for clarity, right order of information given, and missing information; e.g., in "study design" analytical categories are mentioned, but not what the research is all about; the reader might get confused regarding the separation of the research and the meta-research on inclusion/co-research (it gets clearer after reading the whole paper, but it creates some puzzlement in my experience while reading); in order to understand the meta/co-research study, I think more information on the research project is needed

·         L. 140: can you state a bit more explicitly what the research question(s)/context of self-determination was?

Findings

·         L. 569-574: is the statement at the correct place here? It does not really seem to fit

·         what I am wondering at the end of the results section is: apart from interpretation, what was the input you got for the research? was it a modification of research questions?

Discussion

·         l. 648: can you make that more explicit/describe a bit (as you did for the contribution to the interpretation of material by co-researchers with intellectual disabilities, where you gave a very illustrative example)

·         l. 659: is this a new aspect? new paragraph? maybe structure the discussion a bit more with sub-headings if possible?

·         L. 662: respondents: clarify, co-researchers without disability, or both?

·         L. 673: overall, the scientific usefulness/advantages should be made a bit more explicit (is there a publication in preparation showing the results of the actual study the co-researchers participated, with more details on that?)

·         L. 677: what would be a future solution for this dilemma? let independent research group evaluate the meta-research (meta-meta-research)?

·         L. 680: what is referred to here?

·         Starting in line 709, the text reads like a conclusion to me, maybe make such a separate section?

·         L. 720: guidelines: did you already formulate them in detail (in the discussion)? then maybe make a separate section on it

Author Response

Abstract

  • Last sentence is too long, should be split in two.

Thank you for the comment. We agree with it. We deleted a part of this sentence and instead added more information about the findings:

As a result, we have determined several important aspects of inclusive research in the relational perspective involving a co-researcher with a disability and a co-researcher without a disability, and the community perspective. The analyses identified four superordinate themes (building relationships in the research team; opportunities and constraints associated with the implementation of projects; institutional barriers; and the importance of the role of a co-researcher without intellectual disability)

Intro

  • L. 25: suggestion for further elaboration: in which respects (if any) are there different challenges between involving persons with intellectual disability and others, e.g. with patients with physical diseases? is there literature on children co-researchers that might also give some insights that could be useful?

That is a very useful suggestion and definitely exploring it in detail might be of value. In this sentence, however, we did not refer to various groups (e.g. children, minorities, non-heteronormative) as co-researchers. Instead, we noticed that their perspective is not often taken into account.

Method

  • L. 103: provide more information on this project in the beginning/study design section would make sense in my view: when was the project conducted; what was the main aim of the project? 

We have added information about the aim and main activities conducted in the project “Nothing about us without us. Inclusive research involving persons with intellectual disabilities”.

  • L. 107: specify: which premises?

Thank you for the comment. We have added that we meant methodological and ethical premises of inclusive research involving persons with intellectual disabilities.

  • L. 136: how many interviews, how long did they last, where they transcribed? eventually, can you characterise the persons with intellectual disability a bit more (age, degree of disability, etc.)?

We have added necessary information about the project, e.g. the number of focus group and individual interviews. Notably, however, these interviews were not analysed for the purposes of this research paper, so we do not describe the procedure in detail (these results were published and presented at a conference involving persons with disabilities). In this report we analysed the field notes and observation diaries of co-researchers without disabilities.

  • L. 145 “the teams”: specify

We have added necessary information. We meant the two teams of co-researchers embedded in two Polish cities.

  • Overall, the methods section should be checked for clarity, right order of information given, and missing information; e.g., in "study design" analytical categories are mentioned, but not what the research is all about; the reader might get confused regarding the separation of the research and the meta-research on inclusion/co-research (it gets clearer after reading the whole paper, but it creates some puzzlement in my experience while reading); in order to understand the meta/co-research study, I think more information on the research project is needed

Thank you for the comment. We agree with it and tried to clarify it throughout the Method section.

  • L. 140: can you state a bit more explicitly what the research question(s)/context of self-determination was?

We stated what the aim of this project was (lines: 131-133).

Findings

  • L. 569-574: is the statement at the correct place here? It does not really seem to fit

We agree that the statement did not fit perfectly and shortened it by deleting parts that did not exemplify what we wanted to convey. We left some parts however, because we wanted to enhance that co-researchers without disabilities might feel encouraged by co-researchers with disabilities and feel empowered while conducting inclusive research.

  • what I am wondering at the end of the results section is: apart from interpretation, what was the input you got for the research? was it a modification of research questions? 

Actually, the research questions were not modified after completing the project. Rather, in all phases of the project (defining the aim and research questions of the study, creating research tools and infomed consents for participants, conducting research, data analysis and preparing recommendations) co-researchers with disabilities were actively involved. 

Discussion

  • l. 648: can you make that more explicit/describe a bit (as you did for the contribution to the interpretation of material by co-researchers with intellectual disabilities, where you gave a very illustrative example)

Thank you for the comment. We added the passage (lines: 699-704).

Reviewer 2 Report

Comments and Suggestions for Authors

Dear authors,

this is a really interesting research and the results are very important to publish. There are only two general hints:

The procedure should be more described in detail mainly what happened in the eight meetings.

The result chapter and discussion should be structured more with subheadings in due to reader guidance.

Author Response

This is a really interesting research and the results are very important to publish. There are only two general hints:

The procedure should be more described in detail mainly what happened in the eight meetings.

Thank you for the comment. We added some information and a reference to an article in which the training was described in detail.

The result chapter and discussion should be structured more with subheadings in due to reader guidance.

Thank you for the comment. We tried to structure these two parts to make the manuscript clearer. However, having examined other articles published in Social Sciences, we found that there were no subheadings in the Results and Discussion sections. We are ready to add such subheadings provided the Editors allow us to do this.

Reviewer 3 Report

Comments and Suggestions for Authors

The abstract outlines the exploration of inclusive research involving individuals with intellectual disabilities as co-researchers, particularly focusing on the perspective of Polish researchers without disabilities engaged in such projects. Here's a critical evaluation:

Strengths:

·         Focused Objective: The abstract clearly states the aim of analyzing the experiences of researchers without disabilities engaged in inclusive research with individuals with intellectual disabilities.

·         Comprehensive Scope: It covers various aspects such as team relationship building, implementation challenges, institutional barriers, and the role of co-researchers with and without disabilities.

Areas for Improvement:

·         Clarity and Structure: The abstract could benefit from clearer delineation of the analyzed aspects. Each point should be succinctly introduced for better readability.

·         Specific Findings: While it mentions exploring various aspects, it lacks specific findings or insights derived from observations, conversations, or field notes. Adding a brief glimpse into these findings would enhance the abstract's value.

·         Contextual Detailing: The abstract could be more explicit about the context of inclusive research in Poland, detailing any unique challenges or advancements specific to the region.

Recommendations for Enhancement:

·         Concise Summaries: Present each analyzed aspect in a brief, structured manner, providing key observations or conclusions.

·         Illustrate Findings: Include a sentence or two highlighting key findings or insights obtained through the analysis, offering a sneak peek into the article's content.

·         Contextual Contextualization: Offer a concise background or context about inclusive research in Poland, indicating any particular challenges or noteworthy developments.

This introduction provides a comprehensive overview of the landscape of inclusive research involving individuals with intellectual disabilities. Here's a critical assessment:

Strengths:

·         Contextual Clarity: The introduction effectively delineates the evolution of research involving individuals with intellectual disabilities, highlighting the shift from traditional perspectives to inclusive methodologies.

·         Comprehensive Coverage: It covers a wide array of aspects related to inclusive research, including benefits, challenges, ethical considerations, organizational requirements, and strategies for thorough inclusion.

·         Citation Integration: The introduction integrates various citations effectively to support claims and provide a broader background of existing literature in this domain.

Areas for Improvement:

·         Structural Refinement: While the introduction covers various dimensions, the organization could be enhanced for better flow and coherence. It might benefit from clearer subsections to segment different aspects of inclusive research.

·         Specificity in Examples: Including specific case studies or examples of successful inclusive research endeavors could add practical depth to the discussion.

·         Conciseness: Some parts of the introduction might be concise without compromising the depth of information, making it more reader-friendly.

Recommendations for Enhancement:

·         Subsection Organization: Consider breaking down the introduction into subsections, such as 'Evolution of Inclusive Research,' 'Benefits and Challenges,' 'Ethical Considerations,' 'Organizational Requirements,' and 'Strategies for Inclusion.' This would offer better readability and comprehension.

·         Illustrative Examples: Incorporate specific examples or case studies to exemplify successful instances of inclusive research, providing tangible instances of its impact and outcomes.

·         Condensed Information: Review the text to ensure conciseness without losing crucial information. Eliminate redundant or verbose sections to maintain the reader's engagement.

Overall, the introduction effectively sets the stage for discussing inclusive research involving individuals with intellectual disabilities. With improved structural organization, inclusion of specific examples, and concise presentation, it could further engage readers while offering a clear understanding of the topic's nuances.

This method section provides a detailed account of the study design, procedure, and data analysis. Here's a critical evaluation:

Strengths:

·         Clarity in Study Design: The section precisely outlines the objectives, recruitment process, and ethical considerations involved in the study, ensuring a clear understanding of its purpose.

·         Detailed Procedure Description: The step-by-step explanation of collaboration with individuals with intellectual disabilities, including the training sessions and focus interviews, offers comprehensive insights into the research process.

·         Thorough Data Analysis Explanation: The utilization of Braun and Clarke’s content analysis approach is articulated well, elucidating the systematic and rigorous data analysis process.

Areas for Improvement:

·         Elaboration on Challenges Faced: While the section describes the process comprehensively, it could benefit from including potential challenges or limitations encountered during the collaborative research with persons with disabilities.

·         Inclusion of Co-Researchers' Perspectives: It would be valuable to integrate insights or reflections from the co-researchers with intellectual disabilities, providing a more holistic view of the collaborative experience.

Recommendations for Enhancement:

·         Addressing Challenges: Consider incorporating a subsection focusing on any challenges faced during the research process, especially in collaborating with individuals with intellectual disabilities. This addition could offer valuable insights into the complexities and nuances of inclusive research.

·         Reflective Insights: Incorporating excerpts or summaries of the perspectives shared by the co-researchers with disabilities regarding their experience in the research process could enrich the section, offering a more nuanced understanding of the collaborative dynamics.

Overall, while the method section effectively outlines the study design, collaboration process, and data analysis, enhancing it with insights from the co-researchers' perspectives and potential challenges faced during the research would further enrich the comprehensive understanding of conducting inclusive research involving individuals with intellectual disabilities.

The text provides a comprehensive insight into the challenges and opportunities encountered when integrating individuals with intellectual disabilities into research teams. The structure is well-organized, systematically presenting four superordinate themes and their constituent themes, outlining nuances within each category. The specific excerpts and examples cited from the research process offer a vivid understanding of the dynamics within the co-researcher team, shedding light on the complexities faced by individuals with intellectual disabilities and their collaborators.

However, the text could benefit from greater clarity and cohesion in some parts. For instance, while it effectively describes the challenges faced by individuals with disabilities in various aspects of the research process—such as communication difficulties, institutional barriers, and conflicts within the team—there's a need for more explicit connections between these challenges and proposed solutions or strategies for improvement. Providing more explicit recommendations or suggested interventions to address these challenges would enhance the practical value of the findings.

This text discusses the significance of inclusive research involving individuals with disabilities, highlighting the importance of collaboration and the impact of such participation on both disabled and non-disabled co-researchers. It acknowledges the positive outcomes of this collaboration, such as skill development and empowerment for individuals with disabilities, while also addressing the challenges and limitations faced in inclusive research.

The discussion brings attention to critical points regarding the empowerment of co-researchers with disabilities and the need for their genuine involvement in the research process. It emphasizes the potential ethical concerns regarding consent, control over the research process, and the importance of considering the well-being of the co-researchers.

Here's a critical analysis of the text:

Strengths:

·         Emphasis on Collaboration: The text strongly advocates for genuine collaboration and involvement of individuals with disabilities in research, promoting their empowerment and skill development.

·         Ethical Awareness: It raises ethical considerations such as consent, control, and well-being of co-researchers, highlighting the need for ethical practices in inclusive research.

·         Recognizing Benefits and Challenges: Acknowledging the positive impact on co-researchers while discussing the challenges faced in inclusive research demonstrates a balanced perspective.

Areas for Improvement:

·         Clarity in Structure: The discussion lacks a clear structure, making it slightly challenging to follow the arguments and their connections.

·         Need for Specific Examples: While referencing studies strengthens the discussion, providing specific examples or cases would enhance the depth and clarity of the points made.

·         In-Depth Exploration of Challenges: While the challenges are mentioned, a deeper exploration of solutions or strategies to address these challenges would add value to the discussion.

Overall, the text presents a commendable discussion on the significance of inclusive research involving individuals with disabilities, emphasizing collaboration, ethical considerations, and the impact on both disabled and non-disabled researchers. Enhancing clarity, providing specific examples, and delving deeper into challenges and solutions could further enrich the discussion.

Author Response

The abstract outlines the exploration of inclusive research involving individuals with intellectual disabilities as co-researchers, particularly focusing on the perspective of Polish researchers without disabilities engaged in such projects. Here's a critical evaluation:

Strengths:

  • Focused Objective: The abstract clearly states the aim of analyzing the experiences of researchers without disabilities engaged in inclusive research with individuals with intellectual disabilities.
  • Comprehensive Scope: It covers various aspects such as team relationship building, implementation challenges, institutional barriers, and the role of co-researchers with and without disabilities.

Thank you for this positive feedback.

Areas for Improvement:

  • Clarity and Structure: The abstract could benefit from clearer delineation of the analyzed aspects. Each point should be succinctly introduced for better readability.
  • Specific Findings: While it mentions exploring various aspects, it lacks specific findings or insights derived from observations, conversations, or field notes. Adding a brief glimpse into these findings would enhance the abstract's value.
  • Contextual Detailing: The abstract could be more explicit about the context of inclusive research in Poland, detailing any unique challenges or advancements specific to the region.

Thank you for raising these issues. We addressed all of them in the revised Abstract.

Recommendations for Enhancement:

  • Concise Summaries: Present each analyzed aspect in a brief, structured manner, providing key observations or conclusions.
  • Illustrate Findings: Include a sentence or two highlighting key findings or insights obtained through the analysis, offering a sneak peek into the article's content.
  • Contextual Contextualization: Offer a concise background or context about inclusive research in Poland, indicating any particular challenges or noteworthy developments.

We modified the Abstract following all the suggestions.

This introduction provides a comprehensive overview of the landscape of inclusive research involving individuals with intellectual disabilities. Here's a critical assessment:

Strengths:

  • Contextual Clarity: The introduction effectively delineates the evolution of research involving individuals with intellectual disabilities, highlighting the shift from traditional perspectives to inclusive methodologies.
  • Comprehensive Coverage: It covers a wide array of aspects related to inclusive research, including benefits, challenges, ethical considerations, organizational requirements, and strategies for thorough inclusion.
  • Citation Integration: The introduction integrates various citations effectively to support claims and provide a broader background of existing literature in this domain.

Thank you for the positive comments.

Areas for Improvement:

  • Structural Refinement: While the introduction covers various dimensions, the organization could be enhanced for better flow and coherence. It might benefit from clearer subsections to segment different aspects of inclusive research.
  • Specificity in Examples: Including specific case studies or examples of successful inclusive research endeavours could add practical depth to the discussion.
  • Conciseness: Some parts of the introduction might be concise without compromising the depth of information, making it more reader-friendly.

We addressed all the comments raised by the Reviewer.

Recommendations for Enhancement:

  • Subsection Organization: Consider breaking down the introduction into subsections, such as 'Evolution of Inclusive Research,' 'Benefits and Challenges,' 'Ethical Considerations,' 'Organizational Requirements,' and 'Strategies for Inclusion.' This would offer better readability and comprehension.
  • Illustrative Examples: Incorporate specific examples or case studies to exemplify successful instances of inclusive research, providing tangible instances of its impact and outcomes.
  • Condensed Information: Review the text to ensure conciseness without losing crucial information. Eliminate redundant or verbose sections to maintain the reader's engagement.

Overall, the introduction effectively sets the stage for discussing inclusive research involving individuals with intellectual disabilities. With improved structural organization, inclusion of specific examples, and concise presentation, it could further engage readers while offering a clear understanding of the topic's nuances.

Thank you for these useful comments. We revised the Introduction and added subheadings to make the text clearer..

This method section provides a detailed account of the study design, procedure, and data analysis. Here's a critical evaluation:

Strengths:

  • Clarity in Study Design: The section precisely outlines the objectives, recruitment process, and ethical considerations involved in the study, ensuring a clear understanding of its purpose.
  • Detailed Procedure Description: The step-by-step explanation of collaboration with individuals with intellectual disabilities, including the training sessions and focus interviews, offers comprehensive insights into the research process.
  • Thorough Data Analysis Explanation: The utilization of Braun and Clarke’s content analysis approach is articulated well, elucidating the systematic and rigorous data analysis process.

Your feedback is truly appreciated.

Areas for Improvement:

  • Elaboration on Challenges Faced: While the section describes the process comprehensively, it could benefit from including potential challenges or limitations encountered during the collaborative research with persons with disabilities.
  • Inclusion of Co-Researchers' Perspectives: It would be valuable to integrate insights or reflections from the co-researchers with intellectual disabilities, providing a more holistic view of the collaborative experience.

Thank you for the comments. We added information on challenges faced during the project. We also agree that incorporating the perspective of co-researchers with disabilities would enhance the analyses and provide a more holistic view of the experience we all shared and we added a sentence explaining it in the manuscript. In further steps we are planning to conduct interviews with co-researchers with disabilities and prepare a new original article based on their experiences and perspective. 

Recommendations for Enhancement:

  • Addressing Challenges: Consider incorporating a subsection focusing on any challenges faced during the research process, especially in collaborating with individuals with intellectual disabilities. This addition could offer valuable insights into the complexities and nuances of inclusive research.
  • Reflective Insights: Incorporating excerpts or summaries of the perspectives shared by the co-researchers with disabilities regarding their experience in the research process could enrich the section, offering a more nuanced understanding of the collaborative dynamics.

Overall, while the method section effectively outlines the study design, collaboration process, and data analysis, enhancing it with insights from the co-researchers' perspectives and potential challenges faced during the research would further enrich the comprehensive understanding of conducting inclusive research involving individuals with intellectual disabilities.

The text provides a comprehensive insight into the challenges and opportunities encountered when integrating individuals with intellectual disabilities into research teams. The structure is well-organized, systematically presenting four superordinate themes and their constituent themes, outlining nuances within each category. The specific excerpts and examples cited from the research process offer a vivid understanding of the dynamics within the co-researcher team, shedding light on the complexities faced by individuals with intellectual disabilities and their collaborators.

However, the text could benefit from greater clarity and cohesion in some parts. For instance, while it effectively describes the challenges faced by individuals with disabilities in various aspects of the research process—such as communication difficulties, institutional barriers, and conflicts within the team—there's a need for more explicit connections between these challenges and proposed solutions or strategies for improvement. Providing more explicit recommendations or suggested interventions to address these challenges would enhance the practical value of the findings.

This text discusses the significance of inclusive research involving individuals with disabilities, highlighting the importance of collaboration and the impact of such participation on both disabled and non-disabled co-researchers. It acknowledges the positive outcomes of this collaboration, such as skill development and empowerment for individuals with disabilities, while also addressing the challenges and limitations faced in inclusive research.

Thank you for raising these issues. We revised the text to make it clearer and added more information about implications to address the comment.

The discussion brings attention to critical points regarding the empowerment of co-researchers with disabilities and the need for their genuine involvement in the research process. It emphasizes the potential ethical concerns regarding consent, control over the research process, and the importance of considering the well-being of the co-researchers.

Here's a critical analysis of the text:

Strengths:

  • Emphasis on Collaboration: The text strongly advocates for genuine collaboration and involvement of individuals with disabilities in research, promoting their empowerment and skill development.
  • Ethical Awareness: It raises ethical considerations such as consent, control, and well-being of co-researchers, highlighting the need for ethical practices in inclusive research.
  • Recognizing Benefits and Challenges: Acknowledging the positive impact on co-researchers while discussing the challenges faced in inclusive research demonstrates a balanced perspective.

We greatly appreciate the positive comments and feedback.

Areas for Improvement:

  • Clarity in Structure: The discussion lacks a clear structure, making it slightly challenging to follow the arguments and their connections.
  • Need for Specific Examples: While referencing studies strengthens the discussion, providing specific examples or cases would enhance the depth and clarity of the points made.
  • In-Depth Exploration of Challenges: While the challenges are mentioned, a deeper exploration of solutions or strategies to address these challenges would add value to the discussion.

 Thank you for the comments. We hope that the revised version of Discussion is clearer. We also added some examples to enhance the depth of the points we made. Last but not least, more emphasis was put on recommendations based on the findings.

Overall, the text presents a commendable discussion on the significance of inclusive research involving individuals with disabilities, emphasizing collaboration, ethical considerations, and the impact on both disabled and non-disabled researchers. Enhancing clarity, providing specific examples, and delving deeper into challenges and solutions could further enrich the discussion.

Round 2

Reviewer 2 Report

Comments and Suggestions for Authors

Thanks for the response, now it is much clearer.

Author Response

Thank you for this positive feedback.